# Sequencing Analysis of Invasive Carbapenem-Resistant *Klebsiella pneumoniae* Isolates Secondary to Gastrointestinal Colonization

**DOI:** 10.3390/microorganisms13010089

**Published:** 2025-01-05

**Authors:** Gaetano Maugeri, Maddalena Calvo, Dafne Bongiorno, Dalida Bivona, Giuseppe Migliorisi, Grete Francesca Privitera, Guido Scalia, Stefania Stefani

**Affiliations:** 1Microbiology Section, Department of Biomedical and Biotechnological Science, University of Catania, Via Santa Sofia 97, 95123 Catania, Italy; gmaugeri88@gmail.com (G.M.); dafne.bongiorno@unict.it (D.B.); dalidabivona@gmail.com (D.B.); lido@unict.it (G.S.); stefania.stefani@unict.it (S.S.); 2U.O.C. Laboratory Analysis Unit, A.O.U. “Policlinico-San Marco”, Via Santa Sofia 78, 95123 Catania, Italy; 3U.O.C. Laboratory Analysis Unit, A.O. “G.F. Ingrassia”, Corso Calatafimi 1002, 90131 Palermo, Italy; gpp.miglio@gmail.com; 4Department of Clinical and Experimental Medicine, Bioinformatics Unit, University of Catania, Via Santa Sofia 97, 95123 Catania, Italy

**Keywords:** *Klebsiella pneumoniae*, gastrointestinal colonization, systemic infection, resistome, virulome, plasmids, invasive strains

## Abstract

*Klebsiella pneumoniae* represent a common invasive infection etiological agent, whose potential carbapenem-resistance and hypermucoviscosity complicate the patient’s management. Infection development often derives from gastrointestinal colonization; thus, it is fundamental to monitor asymptomatic *K. pneumoniae* colonization through surveillance protocols, especially for intensive care and immunocompromised patients. We described a six-month routine screening protocol from the Policlinico of Catania (Italy), while blood samples were collected from the same patients only in cases of a systemic infection suspicion. All the patients who had dissemination episodes were furtherly investigated through next-generation sequencing, analyzing both colonizing and disseminating strains. This study documents emerging invasive sequence types such as ST101, ST307, and ST395, mainly revealing *blaNDM* or *blaKPC* genes, along with siderophores and hyperproduction capsule markers as virulence factors. Most of the detected factors are presumably related to a specific plasmid content, which are extremely varied and rich. In conclusion, active surveillance through sequencing is essential to enhance awareness of local epidemiology within high-risk multi-drug resistance areas. A random sequencing analysis on the most warning microorganisms could enhance sequence typing (ST) awareness within specific settings, allowing for better prevention control strategies on their eventual persistence or diffusion.

## 1. Introduction

Multi-drug resistant (MDR) Gram-negative bacteria currently represent a significant challenge in hospital settings. Among these microorganisms, the literature data report *Klebsiella pneumoniae* as an invasive infection etiological agent in critical patients. This species often leads to life-threatening diseases due to high virulence expression and increasing beta-lactam resistance rates [1]. Specifically, extended-spectrum beta-lactamases and carbapenemases always complicate antimicrobial treatment [1,2]. Otherwise, some *K. pneumoniae* clones exhibit hypermucoviscosity, siderophores, and other virulence factors contributing to severe respiratory, bloodstream, and nervous system infections [3]. These potential unfortunate outcomes often evolve from an initial *K. pneumoniae* gastro-intestinal colonization. *K. pneumoniae* is an opportunistic pathogen which integrates the gastrointestinal microbiota, occasionally migrating to different anatomical sites and causing hospital-acquired infections (HAIs) [3]. This assumption highlights the importance of strictly monitoring asymptomatic *K. pneumoniae* colonization through surveillance protocols, especially for intensive care and immunocompromised patients [4,5]. European guidelines suggest performing surveillance rectal swabs for *K. pneumoniae* colonization, enhancing the eventual antimicrobial resistance detection. Similar screening protocols significantly determine infection prevention and clinical awareness. Moreover, they limit the adverse impacts on the economic and social spheres [4]. Additionally, previous experimental protocols investigated the multi-drug *K. pneumoniae* genome to accentuate eventual virulence factors or resistome insights [5,6]. The gathered surveillance information about MDR *K. pneumoniae* contributes to enlarging the epidemiological scenario of specific geographic and hospital settings. Identification (ID) and antimicrobial susceptibility testing (AST) inform about the possible *K. pneumoniae* resistance profile, together with particular resistance markers. Furthermore, the eventual third-level sequencing analysis may enhance resistome and virulence patterns [3,5,6,7,8]. The delineation of consistent epidemiological knowledge is essential to intensify infection control procedures, reducing multi-drug-resistant pathogens’ diffusion and inauspicious outcomes in the case of systemic infections. A coherent surveillance strategy is useful, especially within high-risk areas. Regarding our geographic region, *K. pneumoniae* isolates reached 51.6% of carbapenem resistance and 74.9% of third-generation cephalosporins resistance [9]. Despite KPC prevalence, screening protocols recently confirmed the emergence of metallo-β-lactamases-producing *K. pneumoniae* within Southern Italy regions [6,10]. Evidence on NDM production or double-carbapenemases presence makes it urgent to clarify which resistant clones are predominant, enriching the evaluations with virulence factor analysis. With these premises, we performed active epidemiological surveillance through culture-based and sequencing methods to detect extensively drug-resistant (XDR) *K. pneumoniae*. The main aim was to extensively study the proportion between XDR *K. pneumoniae* colonization episodes and transfer contributing to bloodstream infections in the same time interval, trying to document the eventual correspondence between colonizing and infecting strains from the same patient. The high throughput technology defining virulome, resistome, and sequence typing (ST) for both colonizing and infecting *K. pneumoniae* isolates contributed to investigate this possibility, demonstrating the features of *K. pneumoniae*-disseminating isolates. 

## 2. Materials and Methods

### 2.1. Screening Swabs Collection and Culture-Based Procedure

The surveillance protocol placed at the University Hospital Policlinico of Catania (Sicily, Italy) during a six-month (June–December 2023) epidemiological evaluation. The screening involved several critical hospital units, such as surgery, intensive care, hematology, pneumology, internal medicine, and emergency room wards. The recovered patients went through rectal surveillance swabs as a conventional part of the antimicrobial resistance surveillance program. According to this hospital program, a rectal screening for carbapenem-resistant *Enterobacterales* (CRE) is performed on patients’ admission and periodically during the recovery period. The rectal swabs underwent the conventional culture-based method to identify MDR-*K. pneumoniae*.

Culture exams required tryptone soy agar plus 5% of sheep blood (Vakutest Kima, Arzergrande, Italy), which was incubated at 37 °C for 18–24 h, along with meropenem antibiotic disks (Oxoid Thermo Scientific^®^, Waltham, MA, USA) to screen the eventual carbapenem-resistance presence. Specifically, the BD Phoenix^TM^ NID (identification panel) and NMIC-474 (AST panel) were used to complete conventional procedures in the case of meropenem inhibition absence alone.

The current EUCAST guidelines (https://www.eucast.org/fileadmin/src/media/PDFs/EUCAST_files/Breakpoint_tables/v_14.0_Breakpoint_Tables.pdf accessed on the 3 June 2023) guided the minimum inhibitory concentration (MIC) results, reporting the eventual resistance category for coherent MIC values. The carbapenem resistance was confirmed by carbapenemases genes detection using GeneXpert Carba-R cartridges (*Cepheid*). Moreover, blood samples were collected from the same patients in cases of sepsis suspicion. This study did not involve supplementary material or direct intervention on patients, since all the collected samples normally integrated the microbiological diagnosis process.

### 2.2. Molecular In-Depths of Colonizing and Disseminating Strains

All the dissemination episodes were furtherly investigated. Specifically, the rectal and the blood carbapenem-resistant *K. pneumoniae* of the same patient underwent molecular in-depths to reveal a comparison between the strains. The colonization and/or dissemination rates within specific hospital wards were calculated as percentages. Figure 1 summarizes the overall applied protocol.

Sample DNA was extracted by using a QIAamp DNA Mini Kit (Ref. 56304). Library preparation for sequencing on the Illumina platform was performed using a QIAseq^®^ FX DNA Library Core Kit (Ref. 1120146, QIAGEN, Hilden, Germany), starting from 20 ng of genomic DNA sequencing, following the manufacturer’s instructions. The fluorometric Qubit dsDNA HS Assay Kit (Ref. Q32851, Invitrogen, Carlsbad, CA 92008, USA) and the Agilent^®^ High Sensitivity DNA Kit (Ref. 5067-4626) lead to the libraries’ quantification and quality evaluation. Denature and dilute libraries were performed according to the “Denature and Dilute Libraries Guide” protocol by Illumina^®^ (8.5 pM as the loading concentration). Finally, sequencing was conducted with the MiSeq Reagent Kits v3 (Ref. 15043895, Illumina, Inc., 92122, San Diego, CA, USA). The Sample Sheet was created using the Local Run Manager v3 software, following the instructions in the Local Run Manager v3 Software Guide by Illumina [11,12]. Two types of analysis using the QIAGEN CLC Genomics Workbench software, following the User Manual for CLC Microbial Genomics Module v22.0, released on 4 January 2022 (QIAGEN, Aarhus, 8000 Denmark), completed the protocol. The software defined resistance, virulence, and MLST genes [11]. The bioinformatic analysis used TrimGalore (v0.5.0) [13,14] to remove the adapter sequence. Furthermore, Unycicler (v0.4.8) [15] and the Illumina-only assembly modality allowed for the de novo bacterial sequence assembly. Additionally, Kleborate (v2.2.0) and Kaptive [16,17] contributed to investigate *Klebsiella pneumoniae* virulence factors, resistance genes, and capsule loci identification. Prokka (v1.13) [18] was used for bacterial annotation, aligning the output assemblies of the Unycicler with several through *bwa* (0.7.17) to identify punctual mutations in selected genes. TORMES (version 1.3.0) [19] detected metagenome assemblies for antimicrobial resistance screening. For the detection of plasmid replicons, the PlasmidFinder with the bactopia tool was applied, and sequences were correlated to GenBank corresponding plasmid sequences [20,21]. In accordance with default recommendations for plasmid replicon detection, the minimum percentage identity for BLASTn was set to >95% instead of 90%, while target coverage percentage was maintained.

## 3. Results

The surveillance protocol documented a carbapenem-resistant *K. pneumoniae* colonization among 221 patients (46.6%) among the 474 included in the timeframe of the study. The systemic infection episodes occurred within the Intensive Care and Neonatal Intensive Care Units (ICU and NICU), the Haematology Unit, the Emergency Room, and the Neurological Clinic. Overall, this study reported 143 (75.6%) colonized patients within the above-mentioned hospital units, accounting for 7 patients (4.9%) with disseminated infection by the colonizing species. The highest incidence rate appeared within the Intensive Care Units, where two adults and one newborn showed dissemination cases. The Emergency Room accounted for two cases, while the Haematology Unit and the Neurological Clinic presented one single dissemination episode. Table 1 summarizes the numbers and percentages of dissemination episodes within the hospital units.

Table 2 reports phenotypical and genotypical details about the infected patients’ isolates, while Table 3 documents all the antimicrobial susceptibility profile results. First, all these patients reported the same phenotypic MIC results for colonizing and infecting strains. These results exhibited meropenem resistance in all the cases, along with a ceftazidime/avibactam resistance only in the case of metallo-enzymes. The carbapenem-resistance was confirmed by the carbapenemases gene detection using a Carba-R cartridge. This molecular system revealed four patients colonized and infected by KPC-producing *K. pneumoniae*. Otherwise, two patients reported colonizing and infecting NDM-producing strains. Finally, double carbapenemases (NDM and OXA-48) detection was reported in two isolates. The second part of Table 2 shows the specifications about the sequencing analysis, which defined the same ST, resistance, and plasmid patterns for the colonizing and infecting strains of the seven patients.

Furthermore, the analysis confirmed the phenotypic resistance MIC values and the Carba-R results about carbapenemases genes. Table 4 summarizes the virulence markers details on the analyzed strains. Table 4 completes the genomic evidence showing all the details about the plasmids setting within the analyzed isolates. The analysis confirmed the presence of identical plasmid replicons for the colonizing and the infecting strains for each studied patient (Table 5).

Below are some details of the analyzed isolates in each infected patient (Table 2 and Table 4). In regard to the STs, three patients (3,5,7) reported ST395, marked by the k-locus KL2 and genes such as *galF* and *manB* related to specific capsular types, or *rpmA2*, which could correlate with capsule overproduction. Additionally, the ST395 carrying *rpmA2* virulence markers reported a string-test positivity, strengthening the hypothesis of hypermucoviscous phenotypes. The same clone revealed aerobactin (*iutA*, *iucA*-*iucD*), enterobactin (*entB*-*entF*), and yersiniabactin (*irp1-irp2, ybtE-ybtX*) genes (Table 2 and Table 4). The virulome also includes lypo-oligo-polysaccharides (LPS) markers (*wbb* and *wzt*) and phagocytosis inhibition genes (*acr*) (Table 3). In regard to the resistance pattern, the ST395 reveals beta-lactam resistance markers such as *ampH* (an ampC-related gene which simulates a penicillin-binding protein function), NDM-1, OXA-1, TEM-1, and CTXM-15. Finally, mutations in *omp* genes impacted the susceptibility profile (Table 2). All our ST395 strains documented the presence of at least four plasmid replicons, including IncHI1B(pNDM-MAR) (Table 5), which is related to NDM, CTX-M, and OXA-1 markers, along with virulence factors such as *rmpA/A2* and aerobactin (Table 4). All of these resistance and virulence genes appeared within our ST395 isolates. The isolates carry the IncFIIpK91, the ColRNAI, and the IncFIB(K) plasmid replicons (Table 4), among which the last one is related to *bla_OXA-1_*, *bla_CTX-M-15_*, and *bla_NDM-1_* genes. Finally, the ST395 from the patient 5 reported the IncL/M(pOXA-48) plasmid replicon, accounting for five plasmid replicons (Table 5). Three patients documented the ST-101, along with the k-locus KL17 (Table 2). The virulome revealed aerobactin (*iutA*, *iucA*-*iucD*), enterobactin (*entB*-*entF*), yersiniabactin (*irp1*, *irp2*, *ybtS*, *ybtsX*), and phagocytosis inhibition genes (*acr*) (Table 4). Finally, a salmochelin siderophores gene (*iroE*) appeared (Table 4). The resistance profile showed KPC-3 and *omp* mutations, impacting the beta-lactam susceptibility rate (Table 2).

According to Table 2’s summary, patients 1, 2, and 4 showed these characteristics, also exhibiting at least eight identical plasmid replicons. Specifically, all the ST101 were the only STs to present the IncR and the IncFIA (HI1) plasmid replicons (Table 5). Furthermore, ST101 revealed the presence of Col156, Col440I, and Col440II plasmid replicons (Table 5). Only the ST101 from patient 1 presented the IncHI1B(pNDM-MAR), accounting for a total number of nine plasmid replicons (Table 5). Despite this evidence, the strain did not carry OXA-1, CTX-M, or NDM genes, but presents virulence factors such as aerobactin (Table 2). Finally, the ST101 strains revealed the IncFIB(K), IncFIIpKP91, and ColRNAI plasmid replicons (Table 5). The patient 6’s isolates exhibited ST307, along with the k-locus KL102 (Table 2). This ST revealed aerobactin (*iutA* and *iucA*-*iucD*), enterobactin (*entB*-*entF*), LPS (*wbb* and *wzt*), and phagocytosis inhibition genes (*acr*) (Table 4). Furthermore, a salmochelin siderophores gene (*iroE*) emerged (Table 4). ST307 was the only one to show the IncFIB (pQil) plasmid replicon (Table 5). The same ST reported the IncFIB(K), IncFIIpKP91, and the ColRNAI plasmid replicons (Table 5).

## 4. Discussion

*K. pneumoniae* infections currently represent an important healthcare-associated concern due to the microorganism’s capability to cause severe and life-threatening disease. This species often expressed the tendency to acquire new genetic fragments, capturing further virulence and resistance markers [22]. Some virulence factors, such as siderophores, seem to enhance *K. pneumoniae* attitude to invade mucosal districts and organs, concerning clinicians about infection evolution in critically ill patients. Otherwise, carbapenem-resistance genes severely downsize the antimicrobial treatment options for the same species. In these premises, it is essential to strictly monitor fragile patients about *K. pneumoniae* colonization, further investigating the eventual dissemination-related clones.

The present study analyzed the overall clinical features of critical patients colonized by *K. pneumoniae*, whose blood isolation occurred only in seven episodes during the study period. This dissemination rate remarks the previously documented data about K. pneumoniae bloodstream infections in intensive care healthcare settings [23]. Certainly, *K. pneumoniae* invasive infections depend in part on the microorganism-specific characteristics; however, the patient’s clinical conditions significantly contribute to the dissemination events. Our patients documented critical conditions such as respiratory/heart failure, sepsis, and hematological or oncological diseases. These conditions evolved from underlying risk factors such as neutropenia, previous surgical procedures, and antineoplastic treatment. Similar severe conditions often contribute to systemic infection episodes, especially in the case of virulent microorganisms’ previous colonization [24]. Furthermore, multi-drug-resistant microorganisms complicate therapeutic management, enhancing the importance of increasing awareness about Gram-negative antimicrobial resistance diffusion within critical hospital settings. Overall, the main purpose was to demonstrate how virulome and resistome details should follow phenotypical observations in limiting alert microorganisms’ persistence within endemic areas. Our analysis reported interesting details which had a significant impact on local epidemiology and knowledge. First, previously gathered data reported dissemination episodes caused by the conventional ST512/258 *K. pneumoniae* [25]. Despite this premise, our study highlighted the ST101, ST307, and ST395 as invasive etiological agents. The ST101 lineage has been documented as an emerging invasive clone, which significantly persists within hospital settings due to its virulence pattern (long-term persistence episodes). Furthermore, the same clone is related to increased mortality outcomes when compared to different STs [26]. Additionally, the ST307 recently emerged as an invasive clone according to Italian investigations about this ST and blood culture isolates.

Both ST101 and ST307 carry a virulence factor combination which may contribute to environmental persistence and fitness improvements, leading to a consequent replacement of the previously known ST512/258 [26]. Remarkably, the ST395 captured our attention due to several resistance and virulence markers. First, this ST carries a specific capsular type (K2) related to immune escape and hypervirulence. Published data document that K2 serotypes often resist neutrophils and macrophage intervention, probably due to the high sialic acid concentration on their surfaces [27,28]. These immune evasion strategies prolong persistence and virulence. Moreover, the above-mentioned ST also carries rpmA/A2 genes, which correlate with capsule hyperproduction and strain hypermucoviscosity [27,28].

All of these considerations may justify the ST395 persistence within the hospital settings and the fragile hosts. This ST finally reveals the NDM-1 resistance marker, complicating the antimicrobial treatment plan in the case of severe infections.

The disseminated strains belonged to peculiar STs, which did not emerge from colonization episodes during previous active surveillance studies within our hospital setting. Specifically, critical patients revealed colonization by ST46, ST17, and ST35 *K. pneumoniae*, which were never detected in blood culture samples [29]. This evaluation may prove that the ST395, ST101, and ST307 show significantly higher invasive tendencies. The invasive attitude and the reported resistance genes oriented our concern to advanced sequencing investigations about the *K. pneumoniae* plasmids. Our first observation was related to the presence of many different incompatibility groups (approximately 4–9 groups) in the same pairs of isolates. Moreover, all the detected plasmid replicons can be mobile or mobilizable according to the previous literature [30].

Published data documents the presence of the IncFIB(pNDM-Mar) replicon-carrying blaNDM gene in many carbapenem-resistant *K. pneumoniae* all around the world [31]. Henson et al. associated the above-mentioned plasmid replicon with different STs such as ST54 and ST15 [32], enhancing the interest in our ST395 strains carrying IncFIB(pNDM-Mar). All of those strains showed a susceptibility profile extremely related to metallo-β-lactamases production. Additionally, the literature data describes ST147 *K. pneumoniae* carrying bla_OXA-181_, bla_CTX-M-15_, bla_NDM_, and bla_OXA-1_ genes [33]. According to the previously published data, we produced hypothetical observations on our strains. These genes appeared on our ST395 strains, reporting similar resistance patterns. Despite the presence of the IncFIB(pNDM-Mar) replicon, the ST101 from patient 1 did not reveal these resistance markers, showing a susceptibility profile not related to metallo-β-lactamases. The same literature data about the IncFIB(pNDM-Mar) plasmid replicon documents virulence markers, such as aerobactin genes (showed by our ST101 and ST395 strains) and rmpA/A2 genes (showed only by our ST395 strains) [10,33,34]. Other articles report the importance of strictly monitoring some *K. pneumoniae* strains carrying the IncFIB(pNDM-Mar) due to the capability of the pNDM-MAR to diffuse resistance and virulence markers [35]. These data led us to hypothesize a connection between the ST395 dissemination attitude and the detected virulence markers, along with the observation of a metallo-β-lactamases-related susceptibility profile.

Previous data reported the IncR and ColRNAI plasmid replicons in ST395 and ST147 *K. pneumoniae*, along with rpmA/A2, aerobactin, enterobactin, and yersiniabactin genes. All our STs revealed the ColRNAI replicon, which may be related to some of the already cited virulence genes depending on the analyzed ST [36]. Italian authors previously outlined the ColRNAI to multi-drug-resistant *K. pneumoniae* [37]. In addition, the same genetic elements seem to carry conjugation system markers (MOB genes) and toxin-related genes such as paring. These features allow for brilliant genetic information transfers and diffusions [38]. The IncR replicon appeared only within the ST101 isolates, probably reporting aerobactin and yersiniabactin markers. According to the plasmid-related literature, the IncR is a specific ST101 plasmid and always carries SHV genes. As a deduction, blaSHV appears in all of our ST101-IncR plasmid replicons isolates [10]. The scientific literature presented significantly high virulence and mortality rates for the isolates carrying IncR and ColRNAI plasmid replicons, even if it is impossible to collocate specific genes on the analyzed plasmids without complete plasmids sequencing [36]. However, we hypothesized that ST101 showed a dissemination attitude also due to these plasmidic virulence markers, along with carbapenem-resistance MIC values related to β-lactamases genes.

According to published sequencing investigations, the IncL/M(pOXA-48) plasmid reports the blaOXA-48 integration [39,40]. Despite these data, the ST395 reporting this plasmid replicon (patient 5) did not express the same resistance markers, probably due to a genetic fragment loss during transposon integration episodes. The IncFIA (HI1) replicon was reported only for the ST101. The literature data reveal the presence of conjugation-related and genome translocation markers within this plasmid [41]. We did not hypothesize any correlation between this plasmid replicon and phenotypic susceptibility data or virulence elements. The ST307 (patient 6) documented the IncFIB (pQil) plasmid replicon, described as the first to report blaKPC-3 carrying. The KPC marker indeed appeared within our ST307 isolate.

Previous publications documented the same plasmid in *Enterobacterales* isolates within the Southern Italy geographic area [42]. It is an auto-transmittable plasmid whose genomic sequence is similar to the pKPN4 plasmid carrying blaTEM-1 and blaOXA-9 genes [43,44]. Our ST307 isolates may reveal OXA-9 and KPC-3 markers due to the presence of the already-mentioned plasmid. However, further studies will be essential to clarify the full function of OXA-9, whose structure has been described as disrupted by frameshift mutations in the case of KPC-carrying plasmids [45]. ST101 showed the presence of Col156, Col440I, and Col440II plasmid replicons. However, insufficient literature data about these plasmids did not allow for further speculations on their eventual resistance or virulence markers. The same considerations regard the IncFII pKP91, documented in all our isolates.

All the considerations we expressed regarding the plasmid analysis had only an epidemiological value in gathering information about the Inc groups. The limitation of this part is related to the absence of plasmid sequencing and their eventual genomic integration. Third-generation sequencing methodologies should be applied in the future to allow for long-read analyses and investigations of plasmid characteristics. Future studies on plasmids may enhance previously published data about the plasmid *K. pneumoniae*’s antimicrobial resistance and/or virulence genes [46,47]. Remarkably, in silico plasmid multilocus sequence typing (pMLST) could be used as a gold standard plasmid replicon detection method to complete similar analysis [48]. 

Our study emerges within a high-risk epidemiological area for multi-drug resistant *Enterobacterales*. Unfortunately, Europe documented an increase in hypervirulent *K. pneumoniae* isolation and Southern Italy reported significant MDR *Enterobacterales* percentages during recent years [49,50]. On these premises, we aimed to enlarge the microbiological knowledge about MDR *K. pneumoniae* and suspected hypervirulent strains within this species. As a consequence, our results led to an infection control management implementation. Periodical sequencing analysis has been planned to investigate *K. pneumoniae* resistome and virulome after the evidence of multi-drug resistance in the phenotypic antibiogram. Specifically, a random MDR-*K. pneumoniae* sequencing analysis occurs once a quarter within our hospital settings. This protocol highlighted the significant changes in the MDR-*K. pneumoniae* clonal diffusion, focusing the attention on the eventual presence of international warning outbreaks. After the application of these measures, our laboratory and clinical personnel acquired awareness about the local epidemiology and the importance of both investigating virulence and resistance markers.

## 5. Conclusions

In conclusion, active surveillance through sequencing analysis has become essential to enhance awareness about local epidemiology within high-risk multi-drug resistance areas. This approach allowed us to satisfy different aims. For instance, the analysis can confirm the presence of resistance markers, whose detection is already available using the most common diagnostic tools. Additionally, the same analysis may detect new resistance determinants or gene combinations. Finally, this kind of study adds fundamental data about genomes and virulence gene contents. This information cannot emerge from conventional diagnostic methods.

Our study demonstrates how the observation of different susceptibility profiles and strain persistence may guide clinical microbiologists to further surveys about some microbial species. A random sequencing analysis on the most warning microorganisms could enhance ST awareness within specific settings, allowing for better prevention control strategies for their eventual persistence or diffusion. Finally, a plasmid analysis may add informative data on resistance and/or virulence marker transfer within the same healthcare environment, contributing to global knowledge of local STs and possible external gene acquisition. 

## Figures and Tables

**Figure 1 microorganisms-13-00089-f001:**
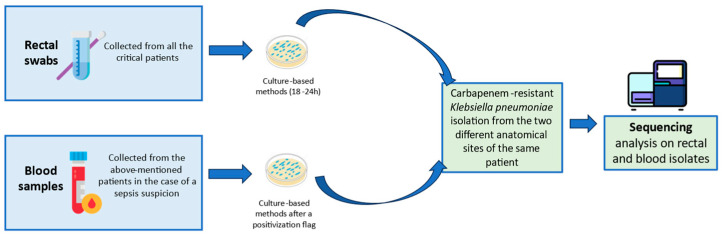
Summary of the applied workflow during active surveillance.

**Table 1 microorganisms-13-00089-t001:** Percentages of the carbapenem-resistant *K. pneumoniae* colonized and infected patients among the different hospital units.

	Number of Colonized Patients	Number of Infected Patients	%
Intensive care units (ICU and NICU)	83	3	3.6%
Haematology Unit	38	1	2.6%
Emergency Room	20	2	10%
Neurological Clinic	2	1	50%
Total	143 *	7	4.9%

* This number refers to the total number of colonized patients only within the hospital wards reporting dissemination episodes during the evaluation.

**Table 2 microorganisms-13-00089-t002:** Phenotypic details, molecular results, and beta-lactams resistome analysis of rectal and blood isolates.

General Information				Phenotypic Susceptibility Data			Molecular Data	Typing Data				Beta-Lactams Resistome Analysis												
Patient	Code	Species	Sample Type	Meropenem MIC (mg/L)	Ceftazidime MIC (mg/L)	Ceftazidime/Avibactam MIC (mg/L)	Carba-R Result	ST	wzi	k-locus	o-locus	*bla_NDM_*	*bla_KPC_*	*bla_OXA_*			*bla_SHV_*				*bla_TEM_*	*bla_CTX-M_*	*bla_ampH_*	*Omp mutations*
1	910091	* K. pneumoniae *	Blood	32	32	2	KPC	101	wzi137	KL17	O1/O2v1	*/*	*KPC-3*	*/*	*/*	*/*	*/*	*/*	*/*	*SHV-212*	*/*	*/*	*ampH*	yes
	885662	* K. pneumoniae *	Rectal swab	32	32	2	KPC	101	wzi137	KL17	O1/O2v1	*/*	*KPC-3*	*/*	*/*	*/*	*/*	*/*	*/*	*SHV-212*	*/*	*/*	*ampH*	yes
2	887213	* K. pneumoniae *	Blood	32	16	2	KPC	101	wzi137	KL17	O1/O2v1	*/*	*KPC-3*	*/*	*/*	*/*	*/*	*/*	*/*	*SHV-212*	*/*	*/*	*ampH*	yes
	887108	* K. pneumoniae *	Rectal swab	32	16	2	KPC	101	wzi137	KL17	O1/O2v1	*/*	*KPC-3*	*/*	*/*	*/*	*/*	*/*	*/*	*SHV-212*	*/*	*/*	*ampH*	yes
3	881005	* K. pneumoniae *	Blood	16	32	32	NDM	395	wzi2	KL2	O1/O2v1	*NDM-1*	*/*	*OXA-1*	*/*	*/*	*/*	*/*	*SHV-187*	*/*	*TEM-181*	*CTX-M-15*	*ampH*	yes
	878504	* K. pneumoniae *	Rectal swab	16	32	32	NDM	395	wzi2	KL2	O1/O2v1	*NDM-1*	*/*	*OXA-1*	*/*	*/*	*/*	*/*	*SHV-187*	*/*	*TEM-181*	*CTX-M-15*	*ampH*	yes
4	911056	* K. pneumoniae *	Blood	16	16	4	KPC	101	wzi137	KL17	O1/O2v1	*/*	*KPC-3*	*/*	*/*	*/*	*/*	*/*	*/*	*SHV-212*	*/*	*/*	*ampH*	yes
	911057	* K. pneumoniae *	Rectal swab	16	16	4	KPC	101	wzi137	KL17	O1/O2v1	*/*	*KPC-3*	*/*	*/*	*/*	*/*	*/*	*/*	*SHV-212*	*/*	*/*	*ampH*	yes
5	886012	* K. pneumoniae *	Blood	16	32	64	NDM/OXA-48	395	wzi2	KL2	O1/O2v1	*NDM-1*	*/*	*OXA-1*	*/*	*OXA-48*	*/*	*/*	*SHV-187*	*/*	*/*	*CTX-M-15*	*ampH*	yes
	878757	* K. pneumoniae *	Rectal swab	16	32	64	NDM/OXA-48	395	wzi2	KL2	O1/O2v1	*NDM-1*	*/*	*OXA-1*	*/*	*OXA-48*	*/*	*/*	*SHV-187*	*/*	*/*	*CTX-M-15*	*ampH*	Yes
6	858616	* K. pneumoniae *	Blood	16	32	2	KPC	307	wzi173	KL102	O1/O2v2	*/*	*KPC-3*	*OXA-1*	*OXA-9*	*/*	*SHV-28*	*/*	*/*	*/*	*TEM-181*	*CTX-M-15*	*ampH*	Yes
	878785	* K. pneumoniae *	Rectal swab	16	32	2	KPC	307	wzi173	KL102	O1/O2v2	*/*	*KPC-3*	*OXA-1*	*OXA-9*	*/*	*SHV-28*	*/*	*/*	*/*	*TEM-181*	*CTX-M-15*	*ampH*	Yes
7	904721	* K. pneumoniae *	Blood	32	16	64	NDM	395	wzi2	KL2	O1/O2v1	*NDM-1*	*/*	*OXA-1*	*/*	*/*	*/*	*/*	*SHV-187*	*/*	*/*	*CTX-M-15*	*ampH*	Yes
	906602	* K. pneumoniae *	Rectal swab	32	16	64	NDM	395	wzi2	KL2	O1/O2v1	*NDM-1*	*/*	*OXA-1*	*/*	*/*	*/*	*/*	*SHV-187*	*/*	*/*	*CTX-M-15*	*ampH*	Yes

**Table 3 microorganisms-13-00089-t003:** Antimicrobial susceptibility testing results for the infected patients.

		Beta-lactams														Fluorochinolone		Aminoglycosides			Colistin	Fosfomycin	Trimethoprim-Sulfamethoxaz
Patient	Sample	AMP	AMC	TZP	FEP	CAZ	CRO	CTX	CXM	AZT	CZA	C/T	MEM	IMI	MEM/VAB	CIP	LEV	AK	CN	TOB	COL	FOS	SXT
1	910091	>8	>32/2	>64/4	>8	>16	>4	>32	>8	>16	8/4	>4/4	>16	>8	≤2/8	>1	>1	>16	>4	>4	≤0.5	≤16	>160
	885662	>8	>16	>64	>16	>32	>4	>32	>8	>16	8	>16	>8	>8	≤2/8	>2	>1	32	>8	>4	≤0.5	≤16	>160
2	887213	>8	>32/2	>64/4	>8	>16	>4	>32	>8	>16	2/4	>4/4	>16	>8	≤2/8	>1	>1	≤4	≤1	≤1	≤0.5	>64	>160
	887108	>8	>16	>64	>16	>32	>4	>32	>8	>16	4	>16	>8	>8	≤2/8	>2	>1	≤1	≤1	1	≤0.5	>64	>160
3	881005	>8	>16	>64	>16	>32	>4	>32	>8	>16	>8	>16	>8	>8	>8/8	>2	>1	32	>8	>8	≤0.5	>64	>160
	878504	>8	>32/2	>64/4	>8	>16	>4	>32	>8	>16	>8/4	>4/4	>16	>8	>8/8	>1	>1	>16	>4	>4	≤0.5	>64	>160
4	911056	>8	>32/2	>64/4	>8	>16	>4	>32	>8	>16	4/4	>4/4	>16	>8	≤2/8	>1	>1	>16	>4	>4	≤0.5	32	>160
	911057	>8	>32/2	>64/4	>8	>16	>4	>32	>8	>16	4	>4/4	>16	>8	≤2/8	>1	>1	>16	>4	>4	≤0.5	32	>160
5	886012	>8	>32/2	>64/4	>8	>16	>4	>32	>8	>16	>8/4	>4/4	16	>8	>8/8	>1	>1	>16	>4	>4	≤0.5	>64	>160
	878757	>8	>16	>64	>16	>32	>4	>32	>8	>16	>8	>16	>8	>8	>8/8	>2	>1	32	>8	>8	≤0.5	>64	>160
6	858616	>8	>32/2	>64/4	>8	>16	>4	>32	>8	>16	2/4	>4/4	16	>8	≤2/8	>1	>1	≤4	≤1	>4	≤0.5	≤16	>160
	878785	>8	>16	>64	>16	>32	>4	>32	>8	>16	1	>16	>8	>8	≤2/8	>2	>1	4	≤1	8	≤0.5	≤16	>160
7	904721	>8	>32/2	>64/4	>8	>16	>4	>32	>8	>16	>8/4	>4/4	>16	>8	>8/8	>1	>1	>16	>4	>4	≤0.5	>64	>160
	906602	>8	>16	>64	>16	>32	>4	>32	>8	>16	>8	>16	>8	>8	>8/8	>2	>1	32	>8	>4	≤0.5	>64	>160

Abbreviations: AMP = Ampicillin; AMC = Amoxicillin/clavulanic acid; TZP = Piperacillin/Tazobactam; FEP = Cefepime; CAZ = Ceftazidime; CRO = Ceftriaxone; CTX = Cefotaxime; CXM = Cefuroxime; AZT = Aztreonam; CZA = Ceftazidim/Avibactam; C/T = Ceftolozane/tazobactam; MEM = Meropenem; IMI = Imipenem; MEM/VAB = Meropenem/Vaborbactam; CIP = Ciprofloxacin; LEV = Levofloxacin; AK = Amikacin; CN = Gentamicin; TOB = Tobramycin; COL = Colistin; FOS = Fosfomycin; SXT = Trimethoprim/sulfametoxazole.

**Table 4 microorganisms-13-00089-t004:** Details about the detected virulence genes in the analyzed strains. The presence of colored marks indicates the positive report of a virulence markers. Specifically, blue marks document aerobactin and phagocytosis inhibition genes, while orange marks indicate LPS and yersiniabactin production genes. Furthermore, green marks are related to enterobactin, salmochelin, and capsule production genes. Finally, light blue marks show the presence of mucoid phenotype and *fimbriae*-related genes.

		Aerobactin					Enterobactin						Capsule Production		Salmochelin	LPS Production				*Mucoid phenotype*	*Type 1-Fimbriae*										*Yersiniabactin*								*Phagocytosis Inhibition*	
ST	Sample Code	*iucA*	*iucB*	*iucC*	*iucD*	*iutA*	*entA*	*entB*	*entC*	*entD*	*entE*	*entF*	*manB*	*galF*	*iroE*	*wbbM*	*wbbN*	*wbbO*	*wzt*	*rmpA2*	*fimA*	*fimB*	*fimC*	*fimD*	*fimE*	*fimF*	*fimG*	*fimH*	*fimI*	*fimK*	*irp1*	*irp2*	*ybtE*	*ybtS*	*ybtT*	*ybtU*	*ybtX*	*ybtA*	*acrA*	*acrB*
101	910091																																							
101	885662																																							
101	887213																																							
101	887108																																							
395	881005																																							
395	878504																																							
101	911056																																							
101	911057																																							
395	886012																																							
395	878757																																							
307	858616																																							
307	878785																																							
395	904721																																							
395	906602																																							

**Table 5 microorganisms-13-00089-t005:** Plasmids analysis results for the dissemination episodes.

	Patient 1		Patient 2		Patient 3		Patient 4		Patient 5		Patient 6		Patient 7	
	Isolate 1	Isolate 2	Isolate 3	Isolate 4	Isolate 5	Isolate 6	Isolate 7	Isolate 8	Isolate 9	Isolate 10	Isolate 11	Isolate 12	Isolate 13	Isolate 14
	910091	885662	887213	887108	881005	878504	911056	911057	886012	878757	858616	878785	904721	906602
	ST 101	ST 101	ST 101	ST 101	ST 395	ST 395	ST 101	ST 101	ST 395	ST 395	ST 307	ST 307	ST 395	ST 395
**PLASMIDS**	IncHI1B(pNDM-MAR) ^a^	IncHI1B(pNDM-MAR) ^a^	-	-	IncHI1B(pNDM-MAR) ^a^	IncHI1B(pNDM-MAR) ^a^	-	-	IncHI1B(pNDM- MAR) ^a^	IncHI1B(pNDM-MAR) ^a^	-	-	IncHI1B(pNDM-MAR) ^a^	IncHI1B(pNDM-MAR) ^a^
	IncFII pKP91 ^b^	IncFII pKP91 ^b^	IncFII pKP91 ^b^	IncFII pKP91 ^b^	IncFII pKP91 ^b^	IncFII pKP91 ^b^	IncFII pKP91 ^b^	IncFII pKP91 ^b^	IncFII pKP91 ^b^	IncFII pKP91 ^b^	IncFII pKP91 ^b^	IncFII pKP91 ^b^	IncFII pKP91 ^b^	IncFII pKP91 ^b^
	IncFIB(K) ^c^	IncFIB(K) ^c^	IncFIB(K) ^c^	IncFIB(K) ^c^	IncFIB(K) ^c^	IncFIB(K) ^c^	IncFIB(K) ^c^	IncFIB(K) ^c^	IncFIB(K) ^c^	IncFIB(K) ^c^	IncFIB(K) ^c^	IncFIB(K) ^c^	IncFIB(K) ^c^	IncFIB(K) ^c^
	IncR ^d^	IncR ^d^	IncR ^d^	IncR ^d^	-	-	IncR ^d^	IncR ^d^	-	-	-	-	-	-
	ColRNAI ^e^	ColRNAI ^e^	ColRNAI ^e^	ColRNAI ^e^	ColRNAI ^e^	ColRNAI ^e^	ColRNAI ^e^	ColRNAI ^e^	ColRNAI ^e^	ColRNAI ^e^	ColRNAI ^e^	ColRNAI ^e^	ColRNAI ^e^	ColRNAI ^e^
	Col156 ^f^	Col156 ^f^	Col156 ^f^	Col156 ^f^	-	-	Col156 ^f^	Col156 ^f^	-	-	-	-	-	-
	IncFIA(HI1) ^g^	IncFIA(HI1) ^g^	IncFIA(HI1) ^g^	IncFIA(HI1) ^g^	-	-	IncFIA(HI1) ^g^	IncFIA(HI1) ^g^	-	-	-	-	-	-
	Col440I ^h^	Col440I ^h^	Col440I ^h^	Col440I ^h^	-	-	Col440I ^h^	Col440I ^h^	-	-	-	-	-	-
	Col440I ^i^	Col440I ^i^	Col440I ^i^	Col440I ^i^	-	-	Col440I ^i^	Col440I ^i^	-	-	-	-	-	-
	-	-	-	-	-	-	-	-	IncL/M(pOXA-48) ^l^	IncL/M(pOXA-48) ^l^	-	-	-	-
	-	-	-	-	-	-	-	-	-	-	IncFIB(pQil) ^m^	IncFIB(pQil) ^m^	-	-

^a^ IncHI1B(pNDM-MAR) (GenBank accession number: JN420336), ^b^ IncFII pKP91(GenBank accession number: CP000966), ^c^ IncFIB(K) Kpn3 (GenBank accession number: JN233704), ^d^ IncR(GenBank accession number: DQ449578), ^e^ ColRNAI(GenBank accession number: DQ298019), ^f^ Col156(GenBank accession number: NC_009781), ^g^ IncFIA (HI1) (GenBank accession number: AF250878), ^h^ Col440I(GenBank accession number: CP023920.1), ^i^ Col440II(GenBank accession number: CP023921.1), ^l^ IncL/M(pOXA-48) (GenBank accession number: JN626286), ^m^ IncFIB(pQil) (GenBank accession number: JN233705).

## Data Availability

The original contributions presented in this study are publicly available. These data can be found at: http://www.ncbi.nlm.nih.gov/bioproject/1125320 (Submission ID: SUB14545389; BioProject ID: PRJNA1125320, Accessed on 18 June 2024).

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
