# Peer review of "Sequencing Analysis of Invasive Carbapenem-Resistant *Klebsiella pneumoniae* Isolates Secondary to Gastrointestinal Colonization"

_microorganisms, 2025, doi:10.3390/microorganisms13010089_

Round 1

Reviewer 1 Report

Comments and Suggestions for Authors

Overall a well-designed and very thorough study.

MDR screening via rectal swabs is commonly performed in ICUs to detect multi-drug resistant strains including K. pneumonia. I'm not sure how feasible or applicable would be such a thorough genetic analysis to such an extent in every ICU/Hematology/ER unit, given the cost over conventional analysis or how it would change the infection control policies already in place - an MDR positive patient is an MDR positive patient and needs to be treated as such!! I would be happy to hear the authors' input on that.

Minor comments: 

Lines 42-44 "These potential unfortunate outcomes evolve from an initial K. pneumoniae gastro-intestinal colonization" Is this always the case, ie is always the GI tract the source of an invasive K. pneumoniae infection?

Lines 152 and 302 The use of the word "globally" implies international data. I understand that the study was performed solely in Italy. Maybe "Overall" might be more appropriate?

Table 3 Please explain the abbreviations of the antibiotics used at the susceptibility test as a footnote.

Author Response

General Comment: Overall, a well-designed and very thorough study. MDR screening via rectal swabs is commonly performed in ICUs to detect multi-drug-resistant strains including K. pneumoniae. I'm not sure how feasible or applicable would be such a thorough genetic analysis to such an extent in every ICU/Hematology/ER unit, given the cost over conventional analysis or how it would change the infection control policies already in place - an MDR positive patient is an MDR positive patient and needs to be treated as such!! I would be happy to hear the authors' input on that.
Answer: Thank you for the observations. Our main purpose was to demonstrate how essential may be to periodically document epidemiological changes about MDR diffusion. We did not propose any extended evaluation, but we think it could be interesting to periodically evaluate MDR microorganisms through advanced technologies to gather definitive resistance and virulence data. As regards the infection control procedures, a better knowledge of the commonly involved microorganisms may lead to a better patients’ management in terms of related personnel, disinfection protocols, and critical patients’ screening during prolonged stays. 

Minor comments

  1. Comment: Lines 42-44 "These potential unfortunate outcomes evolve from an initial K. pneumoniae gastro-intestinal colonization" Is this always the case, i.e. is always the GI tract the source of an invasive K. pneumoniae infection?
    Answer: Thank you for the observation. We decided to add the adverb “often” to improve this sentence meaning.
  2. Comment: Lines 152 and 302 The use of the word "globally" implies international data. I understand that the study was performed solely in Italy. Maybe "Overall" might be more appropriate?
    Answer: Thank you for the comment. We agreed with your suggestion and replaced the term “globally” with the word “overall”.
  3. Comment: Table 3 Please explain the abbreviations of the antibiotics used at the susceptibility test as a footnote.
    Answer: Thank you for the suggestion. We added a footnote.

Please, find all the requested corrections highlighted in yellow.

Reviewer 2 Report

Comments and Suggestions for Authors

The manuscript presents an interesting study in the area of medical microbiology. The paper is generally well structured and written, however, I have some small suggestions for improving the quality of the manuscript.

  1. L70-75 – I suggest re-formulating a testable hypothesis and referring to this hypothesis in the discussion section. Avoid words like „verify”, „analyze” etc. – try to name it more specifically
  2. Please divide the materials and methods section into subsections to make it more readable.
  3. L 120 Did the authors test the quality and quantity of DNA isolates?
  4. Authors presents data only in tables. Maybe it was possible to present some of tchem in more readable graph form?

Author Response

General comments: The manuscript presents an interesting study in the area of medical microbiology. The paper is generally well structured and written, however, I have some small suggestions for improving the quality of the manuscript.
Answer: Thank you for these observations.

  1. Comment: L70-75 – I suggest re-formulating a testable hypothesis and referring to this hypothesis in the discussion section. Avoid words like „verify”, “analyse” etc. – try to name it more specifically
    Answer: Thank you for the suggestion. The paragraph has been revised.
  2. Comment: Please divide the materials and methods section into subsections to make it more readable.
    Answer: We divided the paragraph into two subsections.
  3. Comment: L 120 Did the authors test the quality and quantity of DNA isolates?
    Answer: We added this information within lines 123-126.
  4. Comment: Authors presents data only in tables. Maybe it was possible to present some of them in more readable graph form?
    Answer: We are sorry for the tables’ excessive size. However, it was the only way to include all the collected data, avoiding graphical abstract which may lead to extreme summaries of the detected elements.

Please, find all the requested corrections highlighted in yellow.

Reviewer 3 Report

Comments and Suggestions for Authors

The manuscript entitled “Sequencing analysis of invasive carbapenem-resistant Klebsiella pneumoniae isolates secondary to gastrointestinal colonization” presents a study that documents emerging invasive sequence types, revealing key genes, siderophores, and hyperproduced capsule markers as virulence factors. The impact that the research can generate in future epidemiological studies focused on antimicrobial resistance, especially with carbapenem-resistant K. pneumoniae strains, is highlighted. Below are some recommendations to improve the quality of the manuscript:

·         Although the introduction provides a useful overview of antimicrobial resistance and virulence of K. pneumoniae, it would be valuable to include a clearer discussion on the specific relevance of the study in the global and local context. This could improve the connection between the background presented and the research objectives.

·         The genomic analysis of the colonizing and invasive strains is exhaustive. However, to strengthen the interpretation of the data and its clinical application, the discussion on how the plasmid features and identified virulence factors contribute to the observed resistance profile could be expanded.

·         Although the sequencing methods used are adequate, the absence of long-range sequencing limits the complete understanding of mobile genetic elements and their integration. It is recommended to discuss this limitation and consider future studies using technologies such as Oxford Nanopore or PacBio.

·         For the data on the prevalence of resistant strains and their distribution in different hospital units, significance tests are not included. A more detailed statistical analysis is recommended to strengthen the validity of the conclusions.

·         For the identified virulence factors (such as aerobactin and yersiniabactin) their direct clinical impact on the patients studied could be further discussed. For example, were correlations observed between these factors and the severity of the infection?

Author Response

General comments: The manuscript entitled “Sequencing analysis of invasive carbapenem-resistant Klebsiella pneumoniae isolates secondary to gastrointestinal colonization” presents a study that documents emerging invasive sequence types, revealing key genes, siderophores, and hyperproduced capsule markers as virulence factors. The impact that the research can generate in future epidemiological studies focused on antimicrobial resistance, especially with carbapenem-resistant K. pneumoniae strains, is highlighted. Below are some recommendations to improve the quality of the manuscript.
Answer: Thank you for the observations.

  1. Comment: Although the introduction provides a useful overview of antimicrobial resistance and virulence of K. pneumoniae, it would be valuable to include a clearer discussion on the specific relevance of the study in the global and local context. This could improve the connection between the background presented and the research objectives.
    Answer: Some sentences have been added within the discussion (lines 427-432).
  2. Comment: The genomic analysis of the colonizing and invasive strains is exhaustive. However, to strengthen the interpretation of the data and its clinical application, the discussion on how the plasmid features and identified virulence factors contribute to the observed resistance profile could be expanded.
    Answer: The paragraph has been revised (lines 372-413).
  3. Comment: Although the sequencing methods used are adequate, the absence of long-range sequencing limits the complete understanding of mobile genetic elements and their integration. It is recommended to discuss this limitation and consider future studies using technologies such as Oxford Nanopore or PacBio.
    Answer: We specified the need to perform future studies about this topic within lines 421-422.
  4. Comment: For the data on the prevalence of resistant strains and their distribution in different hospital units, significance tests are not included. A more detailed statistical analysis is recommended to strengthen the validity of the conclusions.
    Answer: We slightly modified the sentences within lines 54-56 and 156-158. Our intention was to only indicate percentages of infection episodes within the different hospital units.  
  5. Comment: For the identified virulence factors (such as aerobactin and yersiniabactin) their direct clinical impact on the patients studied could be further discussed. For example, were correlations observed between these factors and the severity of the infection?
    Answer: Thank you for the interesting observation. We will plan further clinical studies, along with a specific ethical committee authorization about the impact of such virulence factors on patients’ outcomes. We did not evaluate these aspects in the present manuscript because we planned an analysis only on microbial isolates, without any clinical reports collection from patients’ history.

Please, find all the requested corrections highlighted in yellow.

Round 2

Reviewer 1 Report

Comments and Suggestions for Authors

The authors have addressed all my previous suggestions. I am happy for the manuscript to be published in its current form. 

Reviewer 3 Report

Comments and Suggestions for Authors

The authors took into account the recommendations provided to improve the quality of the manuscript.